# Self-Directed Female Migration in Ghana: Health and Wellness of Elderly Family Caregivers Left Behind. An Ethnographic Study

**DOI:** 10.3390/ijerph17218127

**Published:** 2020-11-03

**Authors:** Solina Richter, Kimberly Jarvis, Vida N. Yakong, Patience Aniteye, Helen Vallianatos

**Affiliations:** 1Faculty of Nursing, University of Alberta, Edmonton, AB T6G 1C9, Canada; 2Faculty of Nursing, Memorial University of Newfoundland, St. John’s, NL A1C 5S7, Canada; h02kdj@mun.ca; 3School of Allied Health Sciences, University for Development Studies, Tamale 1350, Ghana; vidayakong@yahoo.ca; 4School of Nursing and Midwifery, University of Ghana, Accra LG 25, Ghana; patienceaniteye@yahoo.co.uk; 5Department of Anthropology, University of Alberta, Edmonton, AB T6G 1C9, Canada; helen.vallianatos@ualberta.ca

**Keywords:** female, migrants, struggles, challenges, left behind, elderly family

## Abstract

Driven by the global economic crisis, families are developing strategies for survival, including self-directed female migration. Female migration has negative and positive impacts on families in rural areas. The purpose of the project was to explore the health and wellness experiences of elderly family caregivers who have female family members who have migrated to improve the status of their families. In this focused ethnographic study, we interviewed elderly family members who had a female family member who migrated outside their community for employment. Participants were enrolled from northern Ghanaian communities known to be economically disadvantaged in comparison to their southern counterparts. All interviews were audio-recorded, transcribed verbatim, and translated into English. Data were analyzed based on thematic content. Majors themes that emerged were reasons for children leaving their families; physical, emotional, and spiritual health; and social and economic struggles. Challenges of family care work undertaken by the elderly in families with emigrated female kin strongly also emerged as a theme. New contextual knowledge was developed about the impact of self-directed female migration on the health and wellness of elderly family caregivers. The information is valuable for the development of culturally appropriate social support and health practices for female migrants and their families.

## 1. Introduction

Driven by the global economic crisis, families are developing a variety of strategies for survival, including self-directed female migration [1]. Women increasingly make decisions about migrating and organizing their work rather than consider family decision-making processes. The feminization of migration is a clear trend within migration in the last few decades. More women migrate to obtain work and support their families [2]. The migration is mostly from rural to urban areas and, in the African context, migration is a family matter that includes sending payments by migrants to their families [3]. When little attention is paid to the social determinants of health (SDOH), it has great consequences for female migrants and their families, specifically the wellness of elderly family members that stay behind. While the financial benefits of female migration have been reported, Dungumaro [3] argues that female migrants have more negative than positive impacts on their families in rural areas. A negative impact may include changes in values and norms that are not always congruent with the traditional values of their communities of origin [2,3].

This project builds upon previous research conducted by a team (Vallianatos, Richter, Ansu-Kyeremeh & Aniteye) in 2014/15, which explored how migration influences the understanding of health and health behaviors by working women who have or have not migrated in Ghana. Insights from this study enriched our understanding of the intersection of migration, gender, and health. Women who migrated had different challenges than migrant men. The harsh environment affected their physical, psychological, and social health and, in particular, safety was a great concern. Our analysis revealed the need to expand the research to northern Ghana, from where most of our migrant participants originated. In low-middle income countries (LMIC), and particularly Ghana, we know little about the effect of migration on the left behind elderly family members. The purpose of the project was to explore the health and wellness experiences of elderly family caregivers (living in northern Ghana) who have female family members who have migrated to improve the financial status of their families. This paper highlights the challenges and opportunities of care work undertaken by the elderly in families with female kin who have migrated.

## 2. Materials and Methods

Our research design was a focused ethnographic study. Focus ethnography concentrates on distinct experiences in a particular culture or subculture [4]. Focused ethnographic research explores participants’ beliefs and practices, viewing them within the context in which they occur rather than aiming to produce findings that can be generalized. The methodology has been used to identify how people from different cultures integrate health beliefs and practices into their lives [5] and to describe the meaning cultures or subcultures ascribe to their experiences [6]. Ethics approval was received from the Human Research Ethics Review Board at the University of Alberta (Pro0071082), Canada. Participants were informed in detail about the research project including the benefits and risks of participation. All research staff signed a confidentiality agreement in which it was explained that all data are confidential and private, and explained their role in maintaining confidentiality. All participants gave oral consent. Participants had the option to withdraw from the study at any time.

We employed convenient and snowball sampling techniques. To be included in the study participants had to be elderly family members age 50 years or above and have or have had a female family member who migrated outside their community for employment. Participants were enrolled from northern Ghanaian communities, which are known to be economically disadvantaged in comparison to their southern counterparts. Data were collected in 2017 over two weeks. We visited the participants at their homes in the local communities. We employed semi-structured interviews and have written extensive field notes. Interviews were conducted with the support of local research assistants who spoke Dagbani, the participants’ native language. All interviews were audio-recorded, transcribed verbatim, and translated into English. A bilingual member from the research team, fluent in English and Dagbani, oversaw the quality of the translation.

Data were analyzed using a word processing software, and were based on thematic content [7]. Each interview was coded to identify concepts of what was communicated. Codes were formulated through a line-by-line analysis of concepts identified in the data. Comparative analysis of codes, and participants’ use of codes, led to the development of subthemes. Themes were developed from both the subthemes that emerged from the data and by comparing to concepts reported in the literature. Rigor was maintained by ensuring the research process was transparent by way of an audit trail, member checking, reflexivity, and ongoing discussion with the research team.

## 3. Results

### 3.1. Demographics

Eighteen participants were interviewed. The ages of the participants ranged between 51 and 90 years, with an average age of 70 years. Seventeen of the participants were women, and one husband-wife pair were interviewed. The participants were all Islamic and earning a living by farming and/or petty trading. Nine participants were widows, one was divorced, and eight were married. Ten of the participants were previously, or at the time of the interviews, in a polygamous marriage. All the participants had female family members who worked away from home as head porters (kayayei), hairdressers, or vendors selling food on the streets and in the markets. All female family members had been working away from home between seven months and seven years.

### 3.2. Major Themes

Majors themes that emerged were reasons for children leaving their families and struggles and challenges of family care work undertaken by the elderly in families with emigrated female kin.

#### 3.2.1. Reasons for Children Leaving Their Families: “She Left Because There Were no Jobs Here”

Participants talked about the reasons their daughters left their families and travelled to the city. The reasons for migrating effected the severity of their health experiences as described in the other themes. The reason for migrating was mainly for work purposes. Most of the participants’ daughters became pregnant at a young age and had to leave school. As it was difficult for them to return to school, most of the girls decided to leave their children with their parents and travel to urban areas to search for work to support the family income.

A mother explained:


*…they are not yet grown. They are still children. It is only this child’s mother who is grown, and she was going to school, but because we do not have money she could not continue. She gave birth and left her baby with me and travelled.*


Another participant added:


*…she just had the baby [at] home, here. She was going to the school but, because of the pregnancy, it was not possible, so she gave birth to the child and we are caring for her. She left her with me and said she will go and see … she said she has regretted so she would be patient and work for money.*


Some participants in this study left home to support other children in the family. School fees were a commonly expressed need, as noted by a participant:


*…because her brother is in school and there is no support, she went there [the city of Accra] so that when school fees come then we will tell her and she brings money for us to pay the fees.*


Most of the participants live in low socioeconomic circumstances and do not have the means to support the dowry needed to make their daughters marriageable. A participant said her daughter had to look for work to buy the kitchen supplies she was expected to have to marry:


*You know, us women if you want to marry and you don’t have bowls [kitchen equipment] you cannot get married, so she decided to go and work and gather her bowls in anticipation of marriage.*


Another participant shared that her health issues prevent her from farming and being self-sustainable. Her daughter left to support her financially:


*I don’t farm. I am not really healthy. I used to prepare shea butter, but since my health isn’t good, I stopped….my unhealthiness is what I live with. My daughter over there gives us food and pocket money and that is what we manage with.*


#### 3.2.2. Struggles and Challenges of Family Care Work Undertaken by the Elderly in Families with Emigrated Female Kin

Participants talked extensively about physical, emotional, spiritual, social, and economic challenges related to their female family members being away from home. Participants shared that children being away contributed to more work around the home and farm and now “*when the day breaks, if I know I have some work to do and no one would be there to help, I just face it.*” They find it difficult to manage with all the daily chores around the house, for example, fetching water and firewood, cleaning, and cooking. There are various responsibilities and lots of hard work for the elderly. Another participant shared:


*I suffer a lot because of her absence. So, I think her stay here at home is much preferred than her stay over there because I cannot even carry water on my head. When I carry water on my head I cannot sleep at night, again when I cook, I cannot sleep at night because of the smoke. Yet, I do all that in her absence because I don’t have anyone to do them for me. And the other daughter is not old enough to do those things.*


Another participant continued and added that she is not physically able to do all the work. She elaborated:


*When they were here and I was strong, they used to work and I will also work and we were all taking care of our responsibilities and needs and by then I did not have any problem, but now they are away and I am no longer strong to be able to work to earn a living.*


Food insecurity was discussed in detail by participants. Their food production is dependent on how much they can harvest and that contributes to feeding the family and finances for other needs. A participant explained:


*…our problem is our farming issues. When it is time to farm and you have enough in your hand to farm, you can cultivate a large scale of land to get enough food that can sustain the family and leave some for sale. If you leave some to sell, at least it can cater for other problems. But, if you don’t have enough food, you don’t even get satisfied, how can you sell some to cater for other problems?*


They continued to say that in these cases their daughters offer support by buying food. A participant described:


*But anytime *[I]* need financial support, she sometimes buys foodstuffs and sends to us.*


Social and economic struggles and challenges experienced focused on their financial need for survival as elaborated on by a participant: “*…our worry is just that we do not have enough [for] our children; they have gone out and since they have gone out and have also left a burden on us, that is our worry.*”

Most participants’ daughters left to contribute to the finances of the family, but family members said, “*that since they have moved, we cannot say there have been changes.*” They do not see many financial benefits of the children working away from home. “*What she sends is not enough to help.*” The financial contribution is dependent on the availability of work in the urban areas, as explained by one mother:


*…there is a bit of change, but it is just that it is not much change, but it is better than how it was. Is only like when she is… she is over there and she has not gotten any [earnings], we will not also get any [money]. It is just not enough, is not much, it doesn’t help.*


Participants shared that it was difficult for their children to sustain themselves while working in the markets and additionally sending money home. An older mother shared that their daughters have to pay for accommodation, food, and other living costs while working in the markets. Their income and how much money they can send home also depends on the availability of work in the markets. She shared:


*…that the child, hmm, that the market has fallen and so they don’t get work like before.*



*…they get but what they used to get they don’t get anymore. *[And]* because she doesn’t get enough is not being able to support. About the finance, since she doesn’t give me, there is no help.*


The daughters that have left often leave their children with their elderly family members and that causes an extra economic burden. A participant shared:


*I am also just suffering and the little I get I have to share with the child she left behind for me. So, it is an extra burden on her having children to take care of. It is just added to you because she is gone.*


Another participant talked about the benefit of her daughter’s contribution to the income of the family. Most of the participants highly valued the education of children. She elaborated:


*…she sent money to the house because when her brother was going to school, she used to send money for the brother to be in school and also her son, her daughter too is in school and because of that, she sends money to take care of her daughter’s school fees because her daughter is going to private school.*


Participants talked about the effect on their emotional and spiritual health as shared: “*I cry in the night because I am thinking of the absence of the children.*” They mentioned worrying about their children being away from home and how they wish they would come home soon to support the family. A participant shared that “*that [they] do not have enough children; they have gone out and since they have gone out and also left a burden on* [them] *that is* [their] *worries.*” Their children’s absence affects their emotional and spiritual health and contributes to physical symptoms such as sleeplessness, as described by an older participant: *“They went because of compelling reasons, so I am worried. Their absence worries me and gives me sleepless nights.”*

Their children’s absence is affecting them *“greatly because if you’re sick and you want to go to hospital, you will need money to go to hospital, you even need someone to carry you to the hospital, but my children are not here to give me money or carry me to hospital and because of that I no longer go to seek for treatment or seek for medication. I will be just in the room and live with God’s favor.”*

Another participant continued*:*


*Now that they (adult children) are not here, *[it]* is not making me happy because it will take a long time and I will not see them and not seeing them and they don’t also have what will make them come so we meet together.*


Participants relied on their religion to manage their emotional health, as described by a participant:


*…if not because of the lack of certain things she would not have gone. Because lack… *[causes]* some difficulties, that is why she has gone. If she were to be around, she would have been helping me in some things. Now that she is away, we only pray that she gets something and comes back, and I will be better again. As she is not there, it worries me so much because if she is there, she will be helping me *[with]* something.*


They “*only pray that* [their daughters] *get something and come back.*” They believe it will all be better if they return.

Participants talked at great length about happiness and what makes them happy and how it contributes to their emotional health. Some remarked about “*financial stability. If you are financially sound, you would be able to take good care of your children without even watching your back. But, if you are there without anything financial* [support]*, you would not be able to look after your kids and that alone can bring you lots of worries.*”

Participants’ sense of happiness is related to physical health and having the means to support their family, as eloquently described by a participant:


*…what I know about happiness is when you wake up and you and your children, they are healthy, and you can get them what they need to eat for the day and tomorrow, then that is happiness. If you are healthy and your children are healthy, then you are happy.*


Happiness was associated with the ability to care for their family members.

## 4. Discussion

Many studies have been conducted on women that are left behind when their husbands become migrant workers [8,9] and the effect on elderly parents when their children migrate [10,11]. Fewer studies have, however, been conducted in the context of low-middle income countries and especially focusing on female migration, leaving their elderly parent(s) behind. In low-middle income countries, the younger women are often the caregivers of their elderly parents, and when they migrate, it has a profound physical, emotional, and social effect on the elderly left behind. The migration of female family members contributes to significant changes in family function and often results in difficulties with kin relationships [12]. Using intersectionality as a framework uncovers the sources of oppression faced by families who decide to have their female family members migrate for economic reasons. Intersectionality is well suited in the health discipline to develop an understanding of the ways in which social locations and identities affect individuals, families, and communities and its effect on caring. Multiple factors intersect and influence the experiences of the elderly left behind, for example, the gender of the adult child who has left, cultural factors (dowry), and oppression (oppressive living arrangement and availability of work that cause stress with the family members left behind).

Our study focused on the wellness experiences of the elderly family members left behind when female members decided to migrate for work purposes. Globally, unemployment, low income, and lack of education have supported younger people’s decisions to migrate. It has become a normal part of younger people’s lives and, more often, younger women are moving, mostly from rural to larger urban areas [13]. In Ghana, the high poverty levels in the northern parts of the country motivate girls and women to migrate to urban areas such as Accra and Kumasi, to engage in what is locally called “kaya business” (head porters) [14]. The need for female members of the family to migrate was influenced by economic needs for survival. It was used as a “temporary option to financially support their families in Northern Ghana” ([15], p. 5). Schooling is free in Ghana, but many children, particularly those living in rural areas, leave school at a young age. Economic necessity forces children, and more so girls, to drop out of school in search of work, to care for younger siblings, and to help with domestic work.

Several societal and cultural norms are pressuring elderly women to fulfill the role of caregivers in the family context [16]. It is important to understand the socioeconomic and cultural context in which these families function. In Africa, and particularly in northern Ghana where this study was conducted, cultural norms are strong and dictate moral conduct and behaviors. Individuality is encouraged and the collective and communal values are deep-seated in the culture [17]. The communality incorporates the function of caring for other persons of the same kinship, clan, or community [18]. Northern Ghana is historically a patriarchal society and women are facing discrimination and inequality in society. In our study, women have, however, independently decided to migrate to find work to support themselves and contribute to the family income. The daughters had children very young and did not complete their schooling. They decided to leave their children with their parents while migrating to work in larger urban areas. This is an acceptable practice in northern Ghana. The concept of family is different from the contemporary Western concept of the nuclear family; “The Dagomba and other northern ethnic groups believe that the child is a gift from God and it is the responsibility of all members of the family to bring up the child” ([19], p. 441). Parents are taking on the caring role of their grandchildren to keep the family ties alive and to ensure that the children have the opportunity to attend school, which might be more difficult if they travel with their mothers [19]. It, however, places a burden on the elderly parents taking on the caregiving role.

Adult children traveled back and forth from northern Ghana as the need arose for financial support. Society assumes that the financial contribution to left behind elderly family members will be beneficial. The common view is that the family member that leaves will contribute to the household income and living standards. Contrary to this assumption, our study shows that elderly parents did not experience it as being beneficial. Dungumaro also describes in his study in rural Tanzania that “migration has not improved household income, it has negatively impacted on migrants’ families in rural areas” ([3], p. 46).

Similarly, in our study, the left behind older family members described physical, socioeconomic, and emotional challenges as a direct impact of their children working away from home. In most African cultures, gender roles are dictated, and responsibilities are assigned to each family member at different stages of their lives [20]. The participants talked about having nobody to support activities normally assigned to women in this society such as cooking, fetching firewood and water, and participating in caring for their crops. Caring for the elderly and supporting them physically and financially is an important task of the female children. The absence of their children additionally affected their mental health, as shared, and also caused sleeplessness and constant worrying. Similarly, some quantitative studies have shown that the out-migration of adult children was highly associated with poor mental health, but the phenomenon was not associated with the physical health of the elderly left behind, as our study portrays [21].

Our findings have policy implications. The health and social care teams in Africa need to consider the values or beliefs of each individual to provide appropriate support needed. Support systems for families need to focus on the negative effects of migration and improving the experiences of the elderly family members that stay behind without adequate social safety nets. Governments must provide better support systems for left-behind elderly family caregivers who participate in childcare and numerous other demanding domestic duties for absent family members, but who may need for care and support themselves [22]. Rural communities should consider developing supportive institutions that can help elderly family members who stay behind to “adapt to the loss of an economically active member or caregiver through migration” ([22], p. 9).

On the macro level, policymakers need to attend to the reasons for migration, which in our case are mainly socioeconomic reasons. More attention is needed to improve the local economy in northern Ghana, for example, Démurger recommends “improving the functioning of labor markets (notably in rural areas, to facilitate the hiring of local labor when a family member migrates), strengthening formal insurance and credit markets, facilitating the transmission of remittances by lowering remitting costs, and increasing access to education and health care” ([11], p. 9). The study has limitations. We mainly interviewed female members that were left behind. It was challenging to recruit male family members; some of them were either deceased or working and not available. We did not include other family members, for example co-wives or siblings. It will be beneficial to conduct a follow up study and interview the family as a unit. The study was conducted in a relative short period. It is recommended to spend more time in these communities and interview participants multiple times to clarify and expand and the data analysis.

## 5. Conclusions

New contextual knowledge was developed about the impact of self-directed female migration on the health and wellness of elderly family caregivers. There is valuable information for the development of culturally appropriate social support and health practices for female migrants and their families applying a social determinant of health framework. A limitation of this study was the unequal sample size of male and female participants. Males hold great authority in northern Ghana; therefore, a greater male representation could have influenced the findings. This is an important area worthy of more research.

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
