# Peer review of "Self-Directed Female Migration in Ghana: Health and Wellness of Elderly Family Caregivers Left Behind. An Ethnographic Study"

_ijerph, 2020, doi:10.3390/ijerph17218127_

Round 1
Reviewer 1 Report
Health and well-being of left-behind family members, especially elderly is an emerging, though less explored issue. Very few qualitative studies are available on this topic. This ethnographic study provides African – Ghanaian context and adds valuable information to the issue.
To increase the visibility, the authors may consider adding the setting and a design to the title. E.g. Self-Directed Female Migration in Ghana: Health and Wellness of Elderly Family Caregivers Left Behind: An Ethnographic Study (OR A Qualitative Study).
When reporting methods, the authors could specify when did data collection take place, how long? In which setting – at household or other? Was any software used to analyse data?
The study title, aim and methodology needs to be in line. Authors may consider to state it more precisely.
Considering the study title and the aim, main outcomes are the health and welfare of LB elderly. However, the results section starts with the major theme: ‘Reasons for children leaving their families’ – which is rather a secondary/additional outcome of the study. I suggest to structure the results section in a way to report all primary outcomes first, and then secondary outcomes. E.g. Primary outcomes: 1. Physical Health 2. Mental - emotional health, happiness. 3. Secondary outcomes – economic wellbeing, motivation for migration, etc. 4. coping strategies: religion, praying. If the information is available, I would suggest to add following details: How participants rate their health and well-being before-after their children migrated? How it changed? do they have an access to healthcare?
Additionally, when reporting results and using quotes, the authors could state more details for each quote - e.g. 79 years old mother of X migrant daughters (who left X years ago) and who takes care of a X children/family member reported ……
The authors stated that 17 out of 18 participants were females. It would be interesting to see how elderly men’s experience is different from women’s experience. The authors may consider explaining the reasons of unequal gender representation, as well as giving more details of a male participant’s experience.
Strength and limitations could be described more explicitly.
Page 1 line 40: I would suggest to give 2-3 examples of a negative and a positive impacts of female migration.
Page 1 line 45-46: The authors state that Insights from a previous study motivated them to extend this research to northern Ghana. The authors also mentioned intersection of migration, gender, and health. More explanation of the motivation and insights would help reader better understand the context. How previous study showed the intersection of migration, gender, and health? What was the main finding of the previous project?
Page 1 line 50-52: Type of migration could be specified. Is the female migration always rural-urban within Ghana, or also includes international migrants?
Page 2 Lines 59-60: More clarification is needed to justify the chosen method. How culture/subculture influences the health beliefs of Left-behind elderly? How health and welfare are defined in this study - as a subjective perception, belief, experience or other?
Page 3 Line 83-85: Seems to be more general remarks how to structure the section; could be removed. Instead, the authors could briefly describe how they structured and analysed the results.
Page 3 line 86-94: It would be easily understandable for readers to present demographics stated in text as a table.
Page 3 Line 93-94: The authors mentioned that duration can be any time from 7 months to 7 years. What is the Emigrants’ frequency of coming back home? Depending on the length of separation, impacts on the LB elderly can be different. The authors may consider integrating these details into results.
Overall, this study is valuable contribution to the LB elderly research.

Author Response
- Revision required
To increase the visibility, the authors may consider adding the setting and a design to the title. E.g. Self-Directed Female Migration in Ghana: Health and Wellness of Elderly Family Caregivers Left Behind: An Ethnographic Study (OR A Qualitative Study).
Response
Thank you, this is very good recommendation. We have changed the title to: Self-Directed Migration in Ghana: health and Wellness of Elderly Family Caregivers Left Behind. An Ethnographic Study.
- Revision required
When reporting methods, the authors could specify when did data collection take place, how long? In which setting – at household or other? Was any software used to analyse data?
Response: We have added the following in formation (Line 71 – 73). Data were collected in 2017 over a period of two week. We visited the participants at their homes in the local communities. We employed semi-structured interviews and written extensive field notes. We added a note on the software used to support the analysis (Line 77).
- Revision required
The study title, aim and methodology needs to be in line. Authors may consider to state it more precisely.
Response: Line 51 – 53. The purposes statement is in line with the title. We have adapted the title.
- Revision required
Considering the study title and the aim, main outcomes are the health and welfare of LB elderly. However, the results section starts with the major theme: ‘Reasons for children leaving their families’ – which is rather a secondary/additional outcome of the study. I suggest to structure the results section in a way to report all primary outcomes first, and then secondary outcomes. E.g. Primary outcomes: 1. Physical Health 2. Mental - emotional health, happiness. 3. Secondary outcomes – economic wellbeing, motivation for migration, etc. 4. coping strategies: religion, praying. If the information is available, I would suggest to add following details: How participants rate their health and well-being before-after their children migrated? How it changed? do they have an access to healthcare?
Response
Again, thanks for the feedback. I agree the reason for migrating is probably a secondary theme but it set up the understanding of some of the findings in the other themes. I prefer to leave it as is. This is a qualitative study and we did not ask participants to rate their health pre- and post the migration of their family members. The ethnographic design does not lend it to this type of analysis.
I have added a clarifying sentence (Lines 105 – 106)
- Revision required
Additionally, when reporting results and using quotes, the authors could state more details for each quote - e.g. 79 years old mother of X migrant daughters (who left X years ago) and who takes care of a X children/family member reported ……
Response
This will interesting to reflect in this manner on the particular quotes, but we have not analysed the data per individual participant but across the group. It will mean we have to go back and re-analysed the data. I am not sure that it will contribute to the quality of the quote and or the understanding of the phenomena.
- Revision required
The authors stated that 17 out of 18 participants were females. It would be interesting to see how elderly men’s experience is different from women’s experience. The authors may consider explaining the reasons of unequal gender representation, as well as giving more details of a male participant’s experience.
Response
This is an interesting remark and ‘yes’ it will be good to go back and interview more male members that were left behind. I have explained the rational in the section where I addressed the limitations; see lines 298 – 303.
- Revision required
Strength and limitations could be described more explicitly.
Response
We expanded on the Limitations (Line 297 – 303).
- Revision required
Page 1 line 40: I would suggest to give 2-3 examples of a negative and a positive impacts of female migration.
Response
We have added positive and negative impacts of female migration. See lines 4- - 43 on page 2
- Revision required
Page 1 line 45-46: The authors state that Insights from a previous study motivated them to extend this research to northern Ghana. The authors also mentioned intersection of migration, gender, and health. More explanation of the motivation and insights would help reader better understand the context. How previous study showed the intersection of migration, gender, and health? What was the main finding of the previous project?
Response
We added two sentences to try to highlight the main findings related to the intersection of gender, migration and health (Line 48 – 49). We tried to keep it very concise.
Reviewer 2 Report
This paper is interesting, especially for anthropologists and social scientists.
I have only minor suggestions:
lines 22-24: the phrase is too long and hard to understand. I suggest to break it into smaller sentences
lines 50-52: almost a repetition of lines 47-50. I suggest to write it in a different way
lines 83-85: I presume these 3 lines are a typo from the article scheme
Author Response
Response to reviewer 2
- Revision required
lines 22-24: the phrase is too long and hard to understand. I suggest to break it into smaller sentences
Response
Line 22- 24The sentence was broken up into two sentences.
- Revision required
lines 50-52: almost a repetition of lines 47-50. I suggest to write it in a different way
Response
Line 52 I have shortened the first question to make it distinct different.
- Revision required
lines 83-85: I presume these 3 lines are a typo from the article scheme
Response
Line 88 Thanks you for catching this error. Yes, it was instruction on how to compile this section. I have deleted it.
Reviewer 3 Report
This paper has a very interesting topic of research. The paper would benefit from some revisions. The authors could elaborate further on the results section and frame it more in the Ghanaian context.
Would it be possible to add more information on the selection criteria for the elderly family members age 50 years and the female migrants. Also move the demographics part to the methods section. This would help to also situate the female migrants better in their region of origin. Is this a selective group?
The results section needs to be analyzed more in-depth and could benefit from some restructuring, in which the findings are better connected to each other. There are too many titles, titles are too long (e.g., 3.2.2.) and this makes it complex to read. The quotes are often too fragmented and need some additional text to demonstrate in-depth analyses. Also make a clearer distinction between the reasons participants give for their health and wellbeing outcomes.
Please provide some more information about the educational system in Ghana and the Ghanaian context (in terms of migration to Accra, school fees, marriage customs, etc.). This could also help to understand the drivers for migration. Provide in the results section also some background information on the participants’ age, profession and living circumstances (e.g. taking care of grandchildren). This is partially done in the discussion section but this needs to be provided earlier in the text.
In the discussion, some new information is provided (e.g., line 258, “In conversations, women shared that their adult children travelled back and forth from northern Ghana as the need arose for financial support” which belongs in the results section.)
Minor comments:
- Abstract: “Female migration has more negative than positive 14 impacts on families in rural areas.” -> try to reformulate in more nuanced words.
- “This section may be divided by subheadings. It should provide a concise and precise description 83 of the experimental results, their interpretation as well as the experimental conclusions that can be drawn.” -> this section is not relevant to add. Also, why would you talk about the “experimental results”?
- Line 118: the word “to” is missing
- Line 227: “Using internationality as a framework uncovers the sources of oppression faced by families who decide to have their female family members migrate for economic reasons.” -> please clarify this internationality framework.
- Not sure whether this sentence, line 277, fits the rest of the text “Particularly, the health care team at large in Africa must ensure they understand the values or beliefs of each individual when relating to them, to provide appropriate support needed.”
Author Response
Response to reviewer 3
Revision required
Would it be possible to add more information on the selection criteria for the elderly family members age 50 years and the female migrants. Also move the demographics part to the methods section. This would help to also situate the female migrants better in their region of origin. Is this a selective group?
Response
The selection criteria of the participants were discussed in line 70 – 74. I believe it is very clear how the participants were recruited. In our discipline it is not common to report on the demographics in the methods section. I prefer to leave at the beginning of the findings section.
Revision required
The results section needs to be analyzed more in-depth and could benefit from some restructuring, in which the findings are better connected to each other. There are too many titles, titles are too long (e.g., 3.2.2.) and this makes it complex to read. The quotes are often too fragmented and need some additional text to demonstrate in-depth analyses. Also make a clearer distinction between the reason’s participants give for their health and wellbeing outcomes.
Response
I have removed some of the subtitle to make the it easier to read; Lines 132, 158 – 159, 185.
Revision required
Please provide some more information about the educational system in Ghana and the Ghanaian context (in terms of migration to Accra, school fees, marriage customs, etc.). This could also help to understand the drivers for migration. Provide in the results section also some background information on the participants’ age, profession and living circumstances (e.g. taking care of grandchildren). This is partially done in the discussion section but this needs to be provided earlier in the text.
Response
I have added two sentences on the schooling in lines 244 – 246.
The demographic section gives the background information of the participants. I am not sure where to more it to introduce it ‘earlier.
Revision required
In the discussion, some new information is provided (e.g., line 258, “In conversations, women shared that their adult children travelled back and forth from northern Ghana as the need arose for financial support” which belongs in the results section.)
Response:
This was an attempt to highlight a key finding shared. I have changed the sentence; see lines 263.
Revision required
Abstract: “Female migration has more negative than positive 14 impacts on families in rural areas.” -> try to reformulate in more nuanced words.
Response
Line 15. I have changed the sentence to “Female migration has negative and positive impacts on families in rural areas”.
Revision required
“This section may be divided by subheadings. It should provide a concise and precise description 83 of the experimental results, their interpretation as well as the experimental conclusions that can be drawn.” -> this section is not relevant to add. Also, why would you talk about the “experimental results”?
Response
I am not sure what the reviewer means here. We did not talk about any experimental result. This is a qualitative study.
Revisions required
Line 118: the word “to” is missing
Response
Corrected – see line 121
Revision required
Line 227: “Using internationality as a framework uncovers the sources of oppression faced by families who decide to have their female family members migrate for economic reasons.” -> please clarify this internationality framework.
Response
Line 232 – 233 I have added a clarifying sentence to make it clearer.
Revision required
Not sure whether this sentence, line 277, fits the rest of the text “Particularly, the health care team at large in Africa must ensure they understand the values or beliefs of each individual when relating to them, to provide appropriate support needed.”
Response
Line 282 - 283The sentence was changes to: The health and social care teams in Africa need to consider the values or beliefs of each individual to provide appropriate support needed” to bring it in line with the rest of the paragraph
Round 2
Reviewer 1 Report
The manuscript has been significantly improved after revision.
The authors considered most of the suggestions and addressed most of the issues. The manuscript is suitable for publication.
Reviewer 3 Report
Thanks for the corrections made.